

**Speciated Atmospheric Mercury during Haze and Non-haze Days in**
**an Inland City in China**
Qianqian Hong[1,3], Zhouqing Xie[1,2,3*], Cheng Liu[1,2,3*], Feiyue Wang[4], Pinhua Xie[2,3],
Hui Kang[1], Jin Xu[2], Jiancheng Wang[1], Fengcheng Wu[2], Pengzhen He[1], Fusheng Mou[2],
Shidong Fan[1], Yunsheng Dong[2], Haicong Zhan[1], Xiawei Yu[1], Xiyuan Chi[1], Jianguo
Liu[2]
1. School of Earth and Space Sciences, University of Science and Technology of
China, Hefei, 230026, China
2. Innovation Center for Excellence in Urban Atmospheric Environment of CAS &
Institute of Urban Environment of CAS, Xiamen, 361021, China
3. Key Lab of Environmental Optics & Technology, Anhui Institute of Optics and
Fine Mechanics, Chinese Academy of Sciences, Hefei, 230031, China
4. Department of Environment and Geography, University of Manitoba, Winnipeg,
Canada
Correspondence author:
zqxie@ustc.edu.cn (Z.Q.X.); chliu81@ustc.edu.cn (C.L.)




**Abstract.** Long-term continuous measurements of speciated atmospheric mercury
were conducted at Hefei, a mid-latitude inland city in east central China, from July
2013 to June 2014. The mean concentrations ($\pm$ standard deviation) of gaseous
elemental mercury (GEM), reactive gaseous mercury (RGM) and particle-bound
mercury (PBM) were $3.95 \pm 1.93$ ng m$^{-3}$, $2.49 \pm 2.41$ pg m$^{-3}$ and $23.3 \pm 90.8$ pg m$^{-3}$,
respectively, during non-haze days, and $4.74 \pm 1.62$ ng m$^{-3}$, $4.32 \pm 8.36$ pg m$^{-3}$ and
$60.2 \pm 131.4$ pg m$^{-3}$, respectively, during haze days. Potential source contribution
function (PSCF) analysis suggested that the atmospheric mercury pollution during
haze days was caused primarily by local mercury emissions, instead of via long-range
mercury transport. In addition, the disadvantageous diffusion during haze days will
also enhance the level of atmospheric mercury. Compared to the GEM and RGM,
change in PBM was more sensitive to the haze pollution. The mean PBM
concentration during haze days was 2.5 times that during non-haze days due to
elevated concentrations of particulate matter. A remarkable seasonal trend in PBM
was observed with concentration decreasing in the following order in response to the
frequency of haze days: autumn, winter, spring, summer. A distinct diurnal
relationship was found between GEM and RGM during haze days, with the peak
values of RGM coinciding with the decline in GEM. Using HgOH as an intermediate
product during GEM oxidation, our results suggest that $NO_2$ aggregation with HgOH
could explain the enhanced production of RGM during the daytime in haze days.
Increasing level of NOx will potentially accelerate the oxidation of GEM despite the
decrease of solar radiation.




## 1. Introduction


Mercury (Hg) is an environmental pollutant that has received much global
attention because of its toxicity and bioaccumulation via the aquatic food chain. The
most important transport pathway of mercury is via the atmosphere (Schroeder and
Munthe, 1998;Lindqvist and Rodhe, 1985). Operationally, atmospheric mercury is
commonly differentiated into three forms: gaseous elemental mercury (GEM),
reactive gaseous mercury (RGM) and particle-bound mercury (PBM). The sum of
these three atmospheric speciated mercury is defined as total atmospheric mercury
(TAM = GEM + RGM + PBM), and the sum of GEM and RGM is known as total
gaseous mercury (TGM = GEM + RGM) (Gustin and Jaffe, 2010;Gustin et al., 2015).
GEM is regarded as the predominant form of atmospheric mercury, accounting for
over 95% of the total. GEM is stable in the atmosphere with a long residence time
(0.5–2 yr) and can be transported at the regional to global scale (Schroeder and
Munthe, 1998;Lindberg et al., 2007). GEM can be oxidized to RGM through
photochemical processes, and further transformed to PBM on aerosol surfaces.
Although GEM is the predominant form of mercury in the air, trace amounts of RGM
and PBM control the mercury scavenged from the atmosphere (Lindberg and Stratton,
1998). RGM and PBM can be readily removed form the air by wet and dry deposition
as a result of their high surface reactivity and water solubility (Lindqvist and Rodhe,
1985). Thus, the chemical transformations between GEM, RGM and PBM will
directly influence the atmospheric lifetime of mercury.
As a result of the rapid industrial development and economic growth of recent
decades, China has become one of the major contributors to anthropogenic mercury
emissions to the environment (Wu et al., 2006;Pacyna et al., 2006;Streets et al., 2005).



Atmospheric mercury emissions from anthropogenic sources in China have been
estimated to be in the range of 500-700 tons/yr, accounting for 25-30% of the total
global anthropogenic mercury emissions (Streets et al., 2005;Wu et al., 2006).
Research into atmospheric mercury in China is therefore critical to the understanding
of mercury cycling at both regional and global scales. Long-term observation of
atmospheric mercury has been conducted in different regions in China, including both
urban and remote areas. TGM concentrations observed in urban and industrial regions
of China were in the range of 2.7–35 ng m$^{-3}$, higher than the values reported for North
America and Europe, and for the adjacent Asian countries such as Korea and Japan
(Stamenkovic et al., 2007;Dommergue et al., 2002;Fang et al., 2009). TGM and PBM
concentrations in remote areas of China were also found to be higher than those
observed in North America and Europe (Fu et al., 2008a;Fu et al., 2008b;Fu et al.,
2012;Liu et al., 2010).
In recent years, haze pollution has become a major concern in China due to its
impacts on visibility, air quality, and climate. It is well known that haze formation is
mainly dependent on the atmospheric relative humidity (RH) and the concentration of
airborne particles (Chen et al., 2003;Sun et al., 2013). Most studies on haze have
focused on the measurements of airborne particulate matter; few examined the
influence of haze on the chemistry of atmospheric mercury, especially PBM. In this
study, we conducted one year synchronous observations of speciated atmospheric
mercury in Hefei, an inland city of China, which experiences frequent haze events.
The comparision of atmospheric mercury under haze days and non-haze days during
the study period allows us to examine the formation and deposition mechanisms of
mercury, as well as their temporal variations.



## 2. Methods

### 2.1 Study site

Continuous measurements of speciated atmospheric mercury were undertaken in Hefei (31°52′ N, 117°17′ E) from July 2013 to June 2014. Hefei, the capital of Anhui Province, is located in east central China, between the Changjiang (Yangtze River) and the Huaihe (Huai River). Hefei has a humid subtropical climate with four distinct seasons: June-August is considered summer, September-November autumn, December-February winter and March-May spring. The prevailing wind is southeasterly in summer and northwesterly in winter. Like many Chinese cities, Hefei has experienced rapid growth in the past 20 years. The total permanent population is about 7.7 million.

The monitoring site was located on the Science Island, a small peninsula on the Dongpu Reservoir in the northwestern outskirts of Hefei (Fig. 1). The sampling and analytical instruments were installed 1.5 m above the rooftop (~ 20 m above the ground) of the main building of Anhui Institute of Optics and Fine Mechanics. Further information about the monitoring site can be found in a previous study (Hu et al., 2014). We chose this area as the monitoring site because it is one of the cleanest areas in Hefei, not adjacent to any direct pollution sources such as power plants, iron and steel works.

### 2.2 Measurements of speciated atmospheric mercury

From July 2013 to June 2014, simultaneous measurements of speciated atmospheric mercury concentrations were performed by an automated Tekran$^{TM}$ mercury speciation system. The system consisted of a Model 2537B mercury analyzer



combined with a Model 1130 RGM unit and a Model 1135 PBM unit. The system was
configured to measure GEM every 5 min., and RGM and PBM every 2 two hr.
The details about the Tekran-based mercury speciation system can be found in
Landis et al. (2002). In general, the automated measurement process can be
summarized as sample collection, thermal desorption and determination. During the
collection period, ambient air was drawn to the system at a flow rate of 10 L/min.
RGM and PBM in the air were captured by a KCl-coated quartz annular denuder in
the 1130 unit and a quartz filter in the 1135 unit, respectively, whereas GEM would
pass through the denuder and filter and be quantified on the Tekran 2537B by
cold-vapor atomic fluorescence spectroscopy (CVAFS). After an hour of sampling, the
1135 quartz filter and the 1130 denuder would be switched to the thermal
decomposition mode at 800 ℃ and 500 ℃, respectively, with the resulting $Hg^0$
quantified by the 2537B unit in the next hour, while the 1135 and 1130 components
were flushed with zero-mercury gas for the next sampling.
The instrument maintenance followed typical protocols used in similar studies
(Landis et al., 2002;Hu et al., 2014). The quartz annular denuder was recoated every
two weeks, the quartz filter was replaced once a month, and the Teflon filter (pore size
0.2 $\mu$m) in the sample inlet was changed every two weeks. Automated recalibration of
the Tekran 2537B was performed every 25 h using an internal mercury permeation
source. No calibration standards were available for RGM and PBM, but the 1σ
precision for RGM and PBM was about 15 % (Landis et al., 2002). The detection
limit in ambient air is about 0.5 ng m$^{-3}$ for GEM (or TGM) at a resolution of 5 min,
and 1 pg m$^{-3}$ for RGM and PBM at a resolution of 2 h (Gustin et al., 2015). Although
the Tekran-based mercury speciation technique has been widely used around the





world, recent studies have shown that the technique does not efficiently collect all
gaseous oxidized mercury and thus may substantially underestimate the concentration
of reactive mercury (Huang et al., 2013;Gustin et al., 2013). Therefore, the RGM
values reported in this study should be considered as the lower limits of gaseous
oxidized mercury in the air (Wang et al., 2014).

**2.3 Ancillary Data**
Standard meteorological measurements including air temperature, air pressure,
RH, wind direction and speed were observed with a 5-min resolution. CO was
measured by an automated infrared carbon monoxide analyzer (Model EC9830T,
Ecotech Inc., Australia), with a detction limit of 40 ppbv. $O_3$ was measured every 5
min by an ozone analyzer (Model EC9810B, Ecotech Inc., Australia); its detection
limit and accuracy are 0.5 ppbv and 0.001 ppm, respectively. $NO_2$ was measured by a
Multi axis differential optical absorption spectroscopy (MAX-DOAS) instrument. The
collected spectra were analyzed using the QDOAS spectral fitting software suite
developed at BIRA-IASB (http://uv-vis.aeronomie.be/software/QDOAS/). We used
the geometric approximation for conversion between slant column densities (SCDs)
and vertical column densities (VCDs) (Ma et al., 2013). $PM_{2.5}$ (particulate matter less
than 2.5 μm in diameter) data are collected from China air quality online analysis
platform (http://www.aqistudy.cn/historydata/index.php). In addition, 24-hr $PM_{10}$
(particulate matter less than 10 μm in diameter) samples were collected on glass-fiber
filters by a high-volume sampler during heavy pollution episodes (from 10 Nov to 9
Dec 2013, n=11). Water-soluble ions in the $PM_{10}$ samples were determined by ion
chromatography (Model ICS-2100, Dionex). In order to identify the potential source





of mercury, NASA's satellite hotspots/fire locations information were obtained from
the Fire Information for Resource Management System (FIRMS)
(https://firms.modaps.eosdis.nasa.gov/firemap/).

**2.4 Potential Sources Contribution Function (PSCF) analysis**
To identify the possible influence of long-range transport on the distribution of
atmospheric mercury in Hefei, we calculated backward trajectories of air masses
using the HYSPLIT (Hybrid Single-particle Lagrangian Integrated Trajectory) model
with the Global Data Assimilation System (GDAS 1°) developed by the National
Oceanic and Atmospheric Administration (NOAA) (*http://www.ready.noaa.gov*)
(Draxler and Hess, 1998). Considering the atmospheric pollutants are mainly
concentrated in the low altitude during heavy pollution days, the trajectory arrival
heights were set at 500 m to represent the boundary layer where atmospheric
pollutants were well mixed. In this study, 5-day back-trajectories were calculated in
ensemble forms which calculate 27 trajectories from the selected starting point ($31°52'$
N, $117°17'$ E) (Fain et al., 2009).
The contributions of other pollution source regions to the atmospheric mercury at
Hefei was identified by the Potential Sources Contribution Function (PSCF) analysis
with the TrajStat software (Wang et al., 2009). PSCF analysis has been shown to be
useful in spatially identifying pollution sources for pollutants with a long lifetime
such as elemental mercury and CO (Xu and Akhtar, 2010). The PSCF values for the
grid cells in the study domain were calculated by counting the trajectory segment
endpoints that terminate within each cell. The number of endpoints that fall in the $ij_{th}$
cell is designated as $N_{ij}$. The number of endpoints for the same cell corresponding to





the atmospheric mercury concentration higher than an arbitrarily set criterion (4 ng
$m^{-3}$ which was the mean GEM concentration during the whole study period) is
defined to be $M_{ij}$. The PSCF value for the $ij_{th}$ cell is then defined as:
$$PSCF_{ij} = \frac{M_{ij}}{N_{ij}} W_{ij} \qquad (2)$$
$W_{ij}$ is an arbitrary weight function to reduce the effect of small values of $N_{ij}$. The
PSCF values were multiplied by $W_{ij}$ to better reflect the uncertainty in the values for
these cells (Polissar et al., 2001). The weight function reduces the PSCF values when
the total number of endpoints in a particular cell is less than 3 times the average value
of the end points per cell:
$$W_{ij} = \begin{cases} 1.0 & N_{ij} \geq 3N_{ave} \\ 0.70 & 3N_{ave} > N_{ij} \geq 1.5N_{ave} \\ 0.40 & 1.5N_{ave} > N_{ij} \geq N_{ave} \\ 0.20 & N_{ave} > N_{ij} \end{cases} \qquad (3)$$

**3.  Results**
We intended to continuously monitor speciated atmospheric mercury
concentration over the course of a year; however, interruptions were inevitable due to
instrument maintenance, which resulted in loss of data for the following four periods:
(1) 25 September to 9 October 2013; (2) 5-14 November 2013; (3) 9-25 February
2014; and (4) 1-14 April 2014. The rest of the data were grouped into haze days and
non-haze days according to the China Meteorological Administration's haze standard
(QX/T 113-2010). Haze days refer to the days when the atmospheric visibility < 10
km and RH < 80% (Duan et al., 2016), and non-haze days refer to clear days with the
atmospheric visibility > 10 km. The visibility and RH information were collected
from    the    weather    history    data    at    the    Luogang    Airport    of    Hefei



(http://www.wunderground.com/). Through the study period of almost a year, a total
of 56 days were identified to be haze days, and 253 days to be non-haze days. All the
times reported herein are local time (UTC + 8 hr).

**3.1  Overall characteristics of speciated atmospheric mercury**
The time series of GEM, RGM and PBM concentrations at the study site
throughout the study period are shown in Fig. 2, and their frequency distributions are
shown in Fig. S1 (in the supporting information). The mean ($\pm$ standard deviation)
GEM, RGM and PBM concentrations during the whole study period were $4.07 \pm 1.91$
ng m$^{-3}$, $3.67 \pm 5.11$ pg m$^{-3}$, and $30.0 \pm 100.3$ pg m$^{-3}$, respectively (Table 1). The GEM
concentrations in different seasons did not differ much. The highest GEM
concentration occurred in autumn ($4.51 \pm 2.10$ ng m$^{-3}$), while the lowest in spring
($3.89 \pm 1.79$ ng m$^{-3}$). RGM concentrations varied greatly during the study period with
much higher concentrations in autumn and the lowest in winter. A similar seasonal
variation in the RGM concentration was observed at a remote site in Mt. Gongga of
southwest China (Fu et al., 2008b). The seasonal trend in PBM was also observed in
Hefei with its concentration decreasing in the following order: autumn > winter >
spring > summer. The mean PBM concentration during the cold season was about 20
times that in summer, similar to the findings from many previous studies in China
(Zhang et al., 2013;Fu et al., 2011;Fu et al., 2008b;Fang et al., 2001).
Comparisons of speciated atmospheric mercury concentrations with other urban
and rural areas in China and a few other countries are shown in Table 2. The mean
GEM concentration at Hefei is slightly higher than that in many remote areas in China
(Fu et al., 2008a;Fu et al., 2008b;Fu et al., 2012;Wan et al., 2009a;Wan et al.,



2009b;Zhang et al., 2015a), but is much lower than those in urban areas of industrial
cities such as Guiyang and Changchun where large point sources of mercury exist
(e.g., non-ferrous metal smelting, coal-fired power plants, and residential coal burning)
(Feng et al., 2004;Fu et al., 2011;Fang et al., 2004). Although Hefei is geographically
close to Shanghai, a mega urban centre in China, it is interesting to note that the TGM
concentration of Shanghai is much lower than that of Hefei. This may be due to the
fact that Shanghai is a coastal city that is influenced more by cleaner marine air
masses (Friedli et al., 2011). Table 2 also shows that the average concentration of
GEM in Hefei is typically more than two folds that in the urban and rural areas in
Europe and North America.

**3.2  Speciated atmospheric mercury during non-haze days**
The frequency distribution of GEM, RGM and PBM for the non-haze period are
shown in Fig. S1 (in blue). The mean concentration of GEM was $3.95 \pm 1.93$ ng m$^{-3}$.
Its distribution was characterized by large fluctuations ranging from 0.2 to 23.8 ng
m$^{-3}$, although more than half of the GEM values were in the narrow range 2-4 ng m$^{-3}$.
The mean concentration of RGM was $2.49 \pm 2.41$ pg m$^{-3}$ with a range of 0.5-33.5 pg
m$^{-3}$, although most of the values were in the range of 1-4 pg m$^{-3}$. High concentrations
of RGM (exceeding 10 pg m$^{-3}$) only accounted for 1.4% of the total data. The mean
RGM concentration at the Hefei site is much smaller than that reported from other
study sites in China (Table 2), but is comparable to the values observed from many
European and North American sites (Brooks et al., 2010;Li et al., 2008;Liu et al.,
2010;Cheng et al., 2014). The mean PBM concentraion at the Hefei site during the
non-haze days was $23.3 \pm 90.8$ pg m$^{-3}$ with an exceptionally large range of 0.5-1827





pg m$^{-3}$. The frequency distribution of PBM showed that high PBM concentrations
(i.e., > 50 pg m$^{-3}$) accounted for 6.4% of the total data. The PBM concentration under
the non-haze condition in Hefei is generally at a similar level to the remote areas, such
as Mt. Gongga, Mt. Waliguan and Shangri-Li in western China.
Diurnal variations of GEM, PBM and RGM concentrations for non-haze days are
shown in Fig. 3. Both GEM and PBM concentrations exhibited similar variations with
elevated concentrations during night. The RGM concentration during the daytime was
slightly higher than that in nighttime, typically peaking between 10:00 and 12:00.

**3.3   Speciated atmospheric mercury during haze days**
Haze pollution mainly occurred in December and January at our monitoring site.
The four major haze pollution periods were identified in grey in Fig. 2. The mean
concentrations of GEM, RGM and PBM during these haze days were 4.74 $\pm$ 1.62 ng
m$^{-3}$, 4.32 $\pm$ 8.36 pg m$^{-3}$ and 60.2 $\pm$ 131.4 pg m$^{-3}$, respectively (Table 1). The frequency
distributions of GEM, RGM and PBM for the haze days are shown in Fig. S1 (in
gray). Comparison of GEM, RGM and PBM concentrations during haze and non-haze
days is shown in Fig. 4. GEM, RGM and PBM concentrations show siginificant
differences between haze and non haze days (p<0.001, t-test). On average, the
concentration of GEM in haze days was 1.2 times that in non-haze days. Similarly, the
concentration of RGM in haze days was about 1-1.7 times those in non-haze days.
The largest impact of haze pollution is however on PBM, with the mean PBM
concentration in haze days about 2.5 times that of non-haze days. High concentrations
of RGM (exceeding 10 pg m$^{-3}$) and PBM concentrations (exceeding 50 pg m$^{-3}$) were





also more frequently observed than in non-haze days, accounting for 5.9% and 25%,
respectively, of the total haze days.
Diurnal variations of GEM, PBM and RGM concentrations for haze days are
shown in Fig. 3. GEM concentrations were higher during night, decreased during
daytime. The opposite pattern was observed for RGM, which showed higher
concentrations during daytime than during night; the duration of the RGM peak also
lasted longer for haze days. On the contrary, the PBM typically peaked just before
sunrise, with the lowest values occurred in the afternoon (14:00-16:00).

**4.   Discussion**
**4.1 Influence of atmospheric mercury emission source**
The statistically significant difference in the GEM concentration between
non-haze days and haze days suggests that haze pollution could directly affect the
concentration of elemental mercury. In order to understand the mercury sources
attribution during haze days, the PSCF model analysis was conducted by using the
TrajStat software. As shown in Fig. 5, the area south to the monitoring site and the
neighboring provinces were the main sources region during haze days. Thus,
atmospheric mercury in haze days were mainly affected by local or regional emission
sources.
The seasonal sources could also be inferred from the PSCF analysis with the
year-round data. Fig. 6(A) showed the overall spatial contribution of mercury
emission sources in China. As Hefei is located in east-central China, its atmospheric
mercury concentration could be affected by both north and south emission sources,
including those from the North China Plain (especially Shandong Provice) and the



neighboring provinces of Henan, Jiangsu, Jiangxi and Hubei. Long-range transport
could also impact the seasonal variations of atmospheric mercury in Hefei. As shown
in Figure 6, in spring, the major contributors of atmospheric mercury to Hefei were
from the southwestern region including the local area and the Jiangxi and Hunan
provinces. In summer, the main contributors were from north of Anhui, as well as
Henan and Jiangxi provinces, and even from the Pearl River Delta region in the far
south. In autumn and winter, the prevalent seasons for haze pollution, the most
important anthropogenic mercury sources to the monitoring site were the local
emissions and those from the neighboring region of Shandong, Henan and the Yangtze
River Delta region. The total mercury emissions from Henan and Shandong provinces
were estimated to be over 50 and 45 tons in 2010, respectively, making them two
largest Hg emitters in China (Zhang et al., 2015b).
GEM and CO normally share anthropogenic emission sources, such as industrial
and domestic coal combustion (Wu et al., 2006). However, they also have their own
sources, vehicles are another kind of dominant sources for CO, while power plants are
another type of mainly sources for GEM. The correlation between the concentrations
of GEM and CO during non-haze and haze days is shown in Fig. S2. The slope of the
trend line represents the Hg/CO ratio. Emissions from power plants typically have a
higher Hg/CO ratio (Wu et al., 2006), whereas biomass burning and residential coal
combustion have a lower Hg/CO ratio due to incomplete combustion (Weiss-Penzias
et al., 2007). The Hg/CO ratios from our study for both non-haze and haze days are in
the range of 0.0003-0.0009 ng m$^{-3}$ ppbv$^{-1}$, similar to the ratio reported for Alaska
biomass burning ($0.0014 \pm 0.0006$ ng m$^{-3}$ ppbv$^{-1}$; Weiss-Penzias et al., 2007),
indicating that biomass burning might have played an important role in mercury



emission in Hefei. This is further supported by the concentration of water-soluble
potassium ($K^+$) in $PM_{10}$. $K^+$ is a typical component of biomass burning aerosol and
has been used as a tracer element for qualitative identification of biomass burning
(Cachier et al., 1991). As shown in Fig. S3, $K^+$ in $PM_{10}$ shows a good correlation with
air quality index (AQI) during the heavy pollution period of Nov-Dec, 2013. In
addition, seven high-GEM events were identified during the whole monitoring period
(Table S1). 5-day backward trajectories for each GEM heavy pollution event for the
time of at maximum GEM concentration are shown in Fig. S4. Air masses with
elevated GEM concentration were mainly from NW, SW and East directions. In
combination with the NASA's satellite hotspots/fire locations information from the
Fire Information for Resource Management System (FIRMS), there were potential
biomass burning occurred in these regions when air masses passed over (Fig. S4,
Events 1-7). Therefore, biomass burning can contribute to the observed higher mercury
concentrations, which not only came from local sources (Events 1 and 4), but could also
be affect by other regions through long-range transport processes (Events 2, 3, 6 and 7).

**4.2 Impacts of meteorological factors for atmospheric mercury during haze days**

Meteorological condition, especially wind direction and speed, could also impact

the atmospheric mercury during haze days. The wind rose for the monitoring site
during the study period is shown in Fig. 7. Easterly and southeasterly winds
represented the prevailing wind directions at the study site. A wind rose diagram of
GEM concentrations above the 90[th] percentile value is shown in Fig. 7B. We found
that 67% of the high GEM concentrations occurred at low wind speed (below 1.5 m
$s^{-1}$); however, wind speed below 1.5 m $s^{-1}$ accounted for only 1.7% of total study.



High RGM and PBM concentrations appear not to be related to high wind speed
(wind speed: 3-5 m s$^{-1}$); only 1.4% and 2.6% of the high RGM and PBM
concentrations were observed under high wind-speed conditions, respectively (Figs.
7C and 7D). In general, most of the high atmospheric mercury levels occurred in the
low wind speed conditions. This slow wind speed condition is not conductive to the
spread of mercury and thus favours the accumulation of atmospheric mercury,
especially during haze days.
Both GEM and PBM concentrations exhibited great variations with elevated
concentration during night or early morning, regardless of the presence of haze. Such a
diurnal variation of GEM and PBM could be related to changes in the height of urban
boundary layer, which is typically low in the morning and night, and high during the
daytime (Yuan et al., 2005;Mao et al., 2006). The maximum PBM concentration
(observed at 6:00) was more than 4 times higher than the minimum value (observed at
16:00) both under non-haze and haze days, and about 76% PBM were removed during
this period (6:00-16:00). However, the reductions of PBM as a result of deposition
during haze days was 62.7 pg m$^{-3}$, which was about 2.4 times that in non-haze days,
suggesting that haze pollution could increase the removal of PBM and thus reduce its
atmospheric lifetime. Although PBM is not the major form of mercury emitted to the
atmosphere, it is crucial in atmospheric mercury transport and removal processes due to
its short atmospheric lifetime. As shown in Fig. 8, the PBM concentration co-varied
with the PM$_{2.5}$ concentration, especially in January when all the four PBM peak events
were associated with increased PM$_{2.5}$ concentrations. The co-variation in February is
weaker, possibly due to the loss of PBM data because of instrument maintenance (see
Section 2.3). Elevated PBM concentrations might be due to the poor diffusion





conditions in cold months and high PM pollution. Although the concentrations of $PM_{2.5}$
were similar from March to June, March showed higher PBM concentrations. This
might be due to higher temperatures in the warmer months which do not favor mercury
adsorption (Otani et al., 1986). These results indicate that both the $PM_{2.5}$ concentration
and temperature may play an important role in the formation of PBM.

**4.3   Chemical process for RGM and the potential oxidation mechanism**

In contrast with the diurnal variations of GEM and PBM, RGM shows different

diurnal trend. The RGM concentration during daytime was slightly higher than at
night. As discussed before in 4.2 section, the diurnal variation of GEM and PBM
could be related to changes in the height of urban boundary layer. So the role of
boundary layer for the enhancement of RGM during daytime would be limited. The
weak correlation (r=0.164, p<0.001) between RGM and CO suggests that regional
anthropogenic sources are not a major source of RGM in the air. As shown in Fig. 9
(haze days), the peak value of RGM coincided well with the lowest value of CO,
suggesting that the production of RGM in haze days does not fully come from
anthropogenic emission sources. Instead, RGM is more likely produced from the in
situ oxidation of GEM; this is supported by the fact that the peak values of RGM
coincided well with the decline of GEM. We found that the RGM concentration did
not change much from 8:00 to 20:00 (local time) during haze days. This indicates that
photochemical reaction between GEM and RGM still takes place during haze days,
but the daytime change in the intensity of solar radiation has been greatly dampened
due to the light-haze interaction.





Various atmospheric oxidants are capable of oxidizing GEM to RGM, including
halogen radicals, ozone, hydroxyl radicals (OH), among others (Holmes et al.,
2010;Wang et al., 2014). Halogen radicals, especially bromine atoms, are believed to
be the the primary oxidant for GEM in the global tropopshere (Holmes et al., 2010).
Unfortunately, we did not measure halogen radicals in this study. Ozone itself is not
an efficient oxidant for GEM oxidation due to low reaction rate (Hall, 1995;Holmes et
al., 2010). In the lower troposphere, ozone is produced from daytime photochemical
reactions involving volatile organic compounds (VOC) and nitrogen oxides ($NO_x$).
Due to fresh emissions of NO from vehicles can react with $O_3$ to form $NO_2$, so ozone
could not fully represent photochemical oxidation processes in urban area (Herndon et
al., 2008). Compared with ozone, odd oxygen ($O_X = O_3 + NO_2$) is a more conserved
tracer of the extent of photochemical processing in the urban atmosphere (Herndon et
al., 2008;Wood et al., 2010). Because of such $NO_2$ concentrations from MAX-DOAS
are available only during the daytime, so we could only use $O_X$ to be a indicator for
GEM oxidation occurred in the daytime. Diurnal variations of GEM, RGM and
Gas-Phase Data ($O_X$, $O_3$, $NO_2$ and CO) concentrations during non-haze and haze days
are shown in Fig. 9. The increase of $O_X$ is consistent with the increase of RGM during
haze days, but seems to lag behind the increase of RGM during non-haze days,
suggesting some oxidation processes might be at work during haze days. During haze
days, both the RGM and $O_X$ reached highest values around 16:00, along with the
lowest value of GEM, indicating the chemical transformation between GEM and
RGM occurred. However, this phenemenon is not found in non-haze days.
The OH radical is also an important oxidant for mercury in the atmosphere.
Previous studies have shown that the major source of OH in the early morning is the



photolysis of HONO, which accumulates in the urban atmospheric boundary layer
during night (Kleffmann et al., 2005). The formation of HgOH as an intermediate
product of the $Hg^0(g)$ + OH oxidation reactions has been proposed by (Sommar et al.,
2001), although HgOH is highly unstable and could decompose back rapidly to $Hg^0$
and OH (Goodsite et al., 2004). It has been proposed that secondary reactants such as
$NO_2$, $HO_2$, RO, $RO_2$, and NO could assist the formation of Hg(II) from the initial
HgOH intermediate, which outcompetes with the decomposition of HgOH (Calvert
and Lindberg, 2005). As an example, we calculated the transformation between GEM
and RGM under the influence of $NO_2$, using the reactions and rate constants shown in
Table S2. As shown in Fig. S5, the production rate of $NO_2HgOH$, $d[NO_2HgOH]/dt$,
increased almost linearly with increasing $NO_2$ under low $NO_2$ concentrations, and
eventually reached a steady state when the $NO_2$ concentration is high enough.
Based on the production rate of $NO_2HgOH$, we can estimate the production of
$NO_2HgOH$ during the 1 hr sampling period when RGM was captured by the
KCl-coated denuder in the Tekran 1130 unit. As discussed earlier, a distinct diurnal
relationship between GEM and RGM was observed both in non-haze and haze days
(Fig. 9). The production of $NO_2HgOH$ and $d[NO_2HgOH]/dt$ corresponding to
different $NO_2$ concentrations is shown in Table 3. With the increase of the $NO_2$
concentration, the contribution of the $NO_2HgOH$ production to RGM will increase. If
the $NO_2$ concentration is within 100 ppbv (from 0 to 100 ppbv), the production of
$NO_2HgOH$ would be in range of 0.058-4.81 pg m$^{-3}$ during the 1h sampling period.
The mean $NO_2$ concentration during haze days is 24.3 ppbv (unit convertion see the
supporting information), which is higher that in non-haze days (17.4 ppbv). The
increments of RGM from sunrise to peak (6:00-12:00) are about 2.14 pg m$^{-3}$ (growth





rate ≈ 0.36 pg m$^{-3}$ h$^{-1}$ ) and 0.41 pg m$^{-3}$ (growth rate ≈ 0.068 pg m$^{-3}$ h$^{-1}$) during haze
days and non-haze days, respectively. As illustrated in Table 3, the level of NO2 in the
urban atmosphere is enough for the production of RGM during the 1h sampling
period. In addition, we found that the NO$_2$ concentrations increased rapidly after
sunrise to reach peak values around noon, consistent with the increases of RGM
during haze days. We thus postulate that NO$_2$ aggregation with HgOH may be a
possible mechanism to explain the enhanced production of RGM during the daytime
over the inland urban air. The NO$_2$ level in urban air might have a more important
influence on the chemical transformations between the GEM and RGM during haze
days, rather than non-haze days. But unfortunately, we can not provide an adequate
description about these oxidation processes. To solve this problem, new laboratory
and mercury model studies on mercury oxidation mechanism are needed.

**5. Summary**
Continuous measurements of speciated atmospheric mercury were conducted at
Hefei, a mid-latitude inland city in central China, from July 2013 to June 2014.
Measurements of other trace gases (e.g. CO, O$_3$, NO$_2$) and meteorological parameters
were employed to better understand the sources and oxidation pathways of
atmospheric mercury. The mean GEM, RGM and PBM concentrations during haze
days were 4.74 ±1.62 ng m$^{-3}$, 4.32 ± 8.36 pg m$^{-3}$ and 60.2 ± 131.4 pg m$^{-3}$, respectively.
Potential source contribution function (PSCF) analysis suggested that the local
mercury emission rather than long-range transport is the most important contributor of
atmospheric mercury pollution during haze days at our monitoring site. The Hg/CO
ratio and NASA's satellite fire locations information indicated that the biomass



burning may plays an important role in mercury emission. Haze pollution has
considerable impact on PBM rather than on GEM and RGM. Both GEM and PBM
concentrations exhibited greatly variations with elevated concentration during night.
The diurnal variations of GEM and PBM might be related to the boundary layer depth;
a lower boundary layer depth in the morning and night could elevate the mercury
concentration. The slow wind speed condition is not conductive to the spread of
mercury and thus favours the accumulation of atmospheric mercury, especially during
haze days. We found that PBM concentrations co-varied with the $PM_{2.5}$ concentration
especially in January when all the four PBM peak events were associated with
increased $PM_{2.5}$ concentrations. In addition, PBM showed a remarkable seasonal
pattern, with higher concentrations in cold seasons and lower in warm seasons.
Elevated PBM concentrations might be due to both the high loadings of particle
matter and disadvantageous diffusion conditions during haze days especially in cold
months. The peaks of RGM were observed around noon, which is probably due to the
higher intensity of solar radiation and photochemical oxidation processes at this time.
Change in the odd oxygen ($O_X = O_3 + NO_2$) concentration is normally appllied to be an
indicator for photochemical reaction. The increase of $O_X$ is consistent with the
increase of RGM during haze days, but seems to lag behind the increase of RGM
during non-haze days, suggesting some oxidation processes might be at work during
haze days. Based on HgOH as an intermediate product, we suggest that $NO_2$
aggregation with HgOH is a potential mechanism to explain the enhanced production
of RGM during the daytime over the inland urban air. The $NO_2$ level in urban air
might have a more important influence on the chemical transformations between the
GEM and RGM during haze days, rather than non-haze days.




**Acknowledgements** This research was supported by grants from the National Basic
Research Program of China (2013CB430000), the National Natural Science
Foundation of China (Project Nos. 91544103,41575021) and the External
Cooperation Program of BIC, CAS (Project No.211134KYSB20130012).

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



**Table 1. Summary of GEM, RGM and PBM concentrations measured in Hefei**
**from July 2013 to June 2014.**

| | GEM (ng m$^{-3}$) | | | RGM (pg m$^{-3}$) | | | PBM (pg m$^{-3}$) | | |
|---|---|---|---|---|---|---|---|---|---|
| | Mean±σ | Range | N | Mean±σ | Range | N | Mean±σ | Range | N |
| Spring | 3.89±1.79 | 0.2-21.3 | 7890 | 4.49±4.22 | 0.5-69.8 | 526 | 8.34±8.97 | 1.6-130.1 | 542 |
| Summer | 4.08±1.99 | 0.3-22.9 | 6050 | 3.66±4.39 | 0.5-45.2 | 511 | 3.61±4.38 | 0.5-41.9 | 570 |
| Autumn | 4.51±2.10 | 0.4-23.8 | 3632 | 5.65±8.93 | 0.5-78.9 | 274 | 59.9±153.5 | 0.5-1615 | 339 |
| Winter | 4.05±1.81 | 0.9-12.2 | 6381 | 2.59±2.58 | 0.5-9.5 | 541 | 56.1±134.9 | 0.5-1827 | 639 |
| Total | 4.07±1.91 | 0.2-23.8 | 23953 | 3.67±5.11 | 0.5-78.9 | 1852 | 30.02±100.3 | 0.5-1827 | 2090 |
| Non-haze | 3.95±1.93 | 0.2-23.8 | 20345 | 2.49±2.41 | 0.5-33.5 | 1508 | 23.3±90.76 | 0.5-1827 | 1708 |
| Haze | 4.74±1.62 | 2.1-16.5 | 3608 | 4.32±8.36 | 0.5-78.9 | 344 | 60.2±131.4 | 1.6-1615 | 382 |





**Table 2. Speciated atmospheric mercury concentrations in Hefei and other urban**
**and rural areas.**


| Location | Classification | Time | TGM (ng m$^{-3}$) | GEM (ng m$^{-3}$) | RGM (pg m$^{-3}$) | PBM (pg m$^{-3}$) | Reference |
|---|---|---|---|---|---|---|---|
| Hefei | Suburb | Jul 2013-Jun 2014 | 4.1 | 4.07 | 3.67 | 30 | This study |
| Hefei | Suburb | Feb-May 2009 | 2.53 | - | - | - | Hu et al. (2014) |
| Beijing | Rural | Dec 2008-Nov 2009 | 3.23 | 3.22 | 10.1 | 98.2 | Zhang et al. (2013) |
| Shanghai | Urban | Aug-Sep 2009 | 2.7 | - | - | - | Friedli et al. (2011) |
| Nanjing | Urban | Jan-Dec 2011 | 7.9 | - | - | - | Zhu et al. (2012) |
| Guiyang | Urban | Nov 2001-Nov 2002 | 8.4 | - | - | - | Feng et al. (2004) |
| Guiyang | Urban | Aug-Dec 2009 | - | 9.72 | 35.7 | 368 | Fu et al. (2011) |
| Changchun | Urban | Jul 1999-Jan 2000 | 18.4 | - | - | 276 | Fang et al. (2004) |
| Changchun | Suburb | Jul 1999-Jan 2000 | 11.7 | - | - | 109 | Fang et al. (2004) |
| Mt.Changbai | Remote | Aug 2005-Jul 2006 | 3.58 | - | 65 | 77 | Wan et al. (2009a, b) |
| Mt.Gongga | Remote | May 2005-July 2006 | 3.98 | - | 6.2 | 30.7 | Fu et al. (2008a, b) |
| Mt.Waliguan | Remote | Sep 2007-Aug 2008 | 1.98 | - | 7.4 | 19.4 | Fu et al. (2012a) |
| Mt.Leigong | Remote | May 2008-May 2009 | 2.8 | - | - | - | Fu et al. (2010) |
| Shangri-La | Remote | Nov 2009-Nov 2010 | 2.55 | - | 8.22 | 38.82 | Zhang (2015) |
| Detroit, USA | Urban | Jan-Dec 2004 | - | 2.5 | 15.5 | 18.1 | Liu et al. (2010) |
| Dexter, USA | Rural | Jan-Dec 2004 | - | 1.6 | 3.8 | 6.1 | Liu et al. (2010) |
| Houston, USA | Urban | Aug-Oct 2006 | - | 1.66 | 6.9 | 2.5 | Brooks et al. (2010) |
| Göteborg, Sweden | Urban | Feb-Mar 2005 | - | 1.96 | 2.53 | 12.5 | Li et al. (2008) |
| Nova Scotia, Canada | Urban | Jan 2010- Dec 2011 | | 1.67 | 2.07 | 2.32 | Cheng et al. (2014) |
| Northern Hemisphere background value | | | | 1.5-1.7 | | | Lindberg et al. (2007) |





**Table 3. The production of NO$_2$HgOH and d[NO$_2$HgOH]/dt at different NO$_2$**
**concentrations**

| NO2 (ppbv) | 10 | 20 | 30 | 40 | 50 | 60 | 70 | 80 | 90 | 100 |
|---|---|---|---|---|---|---|---|---|---|---|
| d(NO$_2$HgOH)/dt (molecule cm$^{-3}$ s$^{-1}$) | 0.36 | 0.71 | 1.04 | 1.37 | 1.68 | 1.99 | 2.28 | 2.56 | 2.83 | 3.10 |
| NO2HgOH (pg m$^{-3}$, 1hr) | 0.56 | 1.10 | 1.63 | 2.13 | 2.61 | 3.08 | 3.54 | 3.97 | 4.40 | 4.81 |





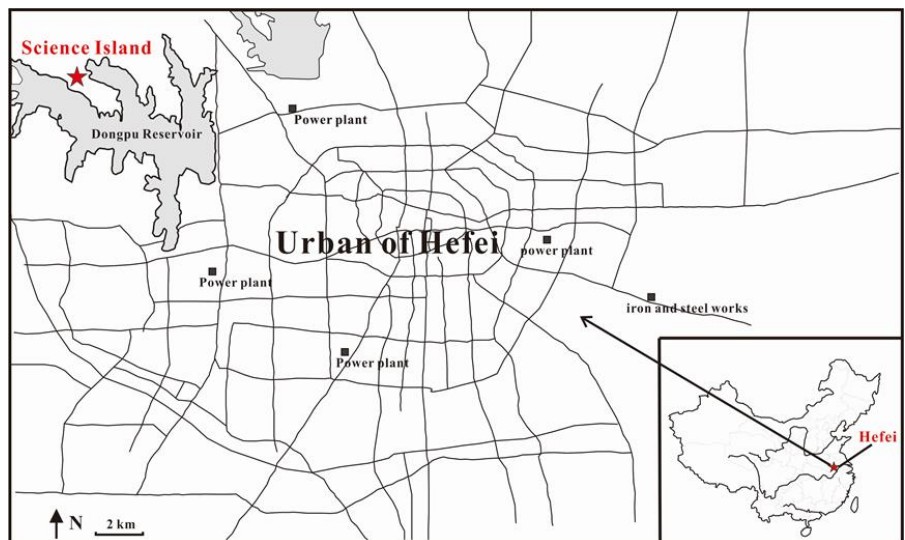



**Fig. 1.** Location of the monitoring site in Hefei, China.




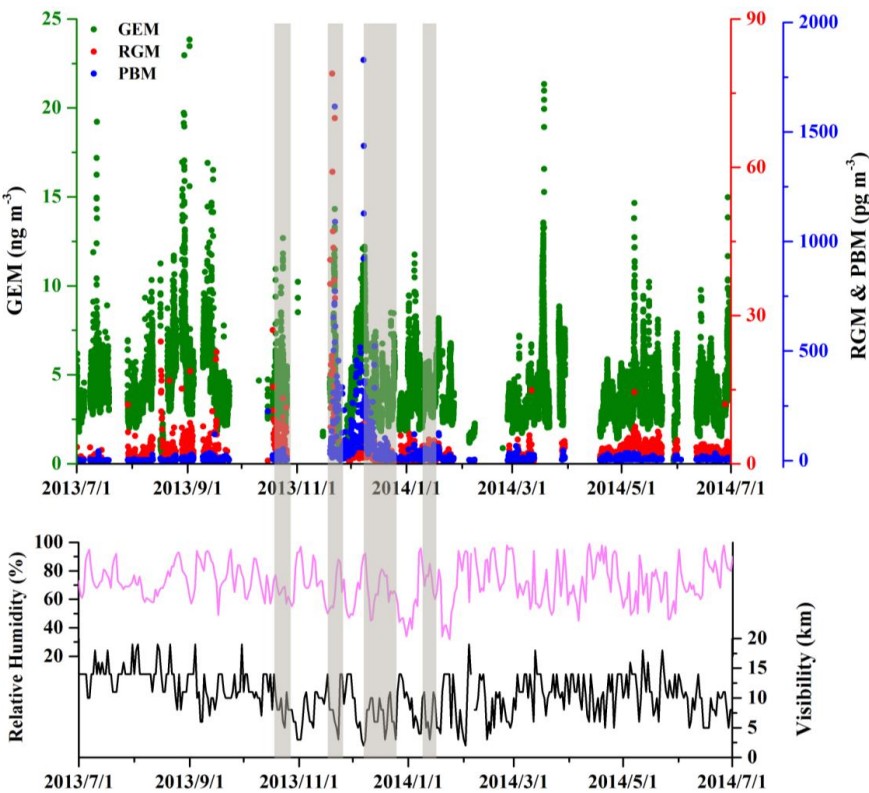


**Fig. 2.** Time series of GEM, RGM and PBM concentrations, along with visibility,

relatively humidity, at the monitoring site in Hefei from July 2013 to June 2014. The

GEM data were at a 5-min resolution, and the RGM and PBM data were two-hour

averages. The gray columns show the major haze pollution episodes occurred during

the study period.





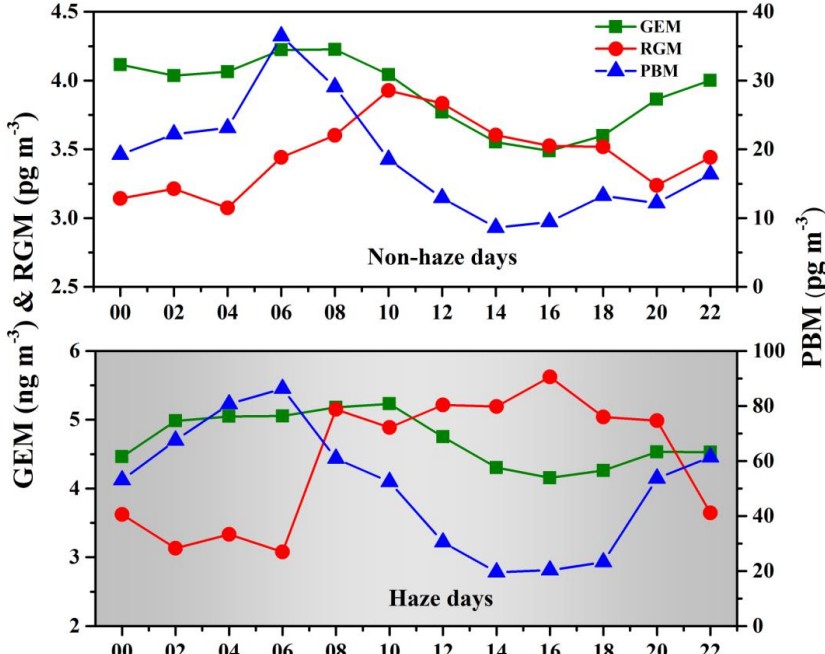


**Fig. 3.** Diurnal trends of GEM, RGM and PBM concentrations in Hefei during
non-haze and haze days (Local time = UTC + 8 hr). The data were two-hour averages.



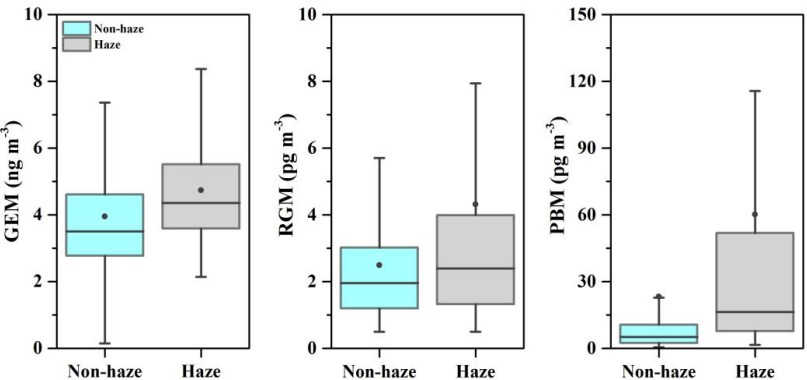


**Fig. 4.** GEM, RGM and PBM concentrations during non-haze and haze days. The
GEM data were at a 5-min resolution, the RGM and PBM data were two-hour
averages.



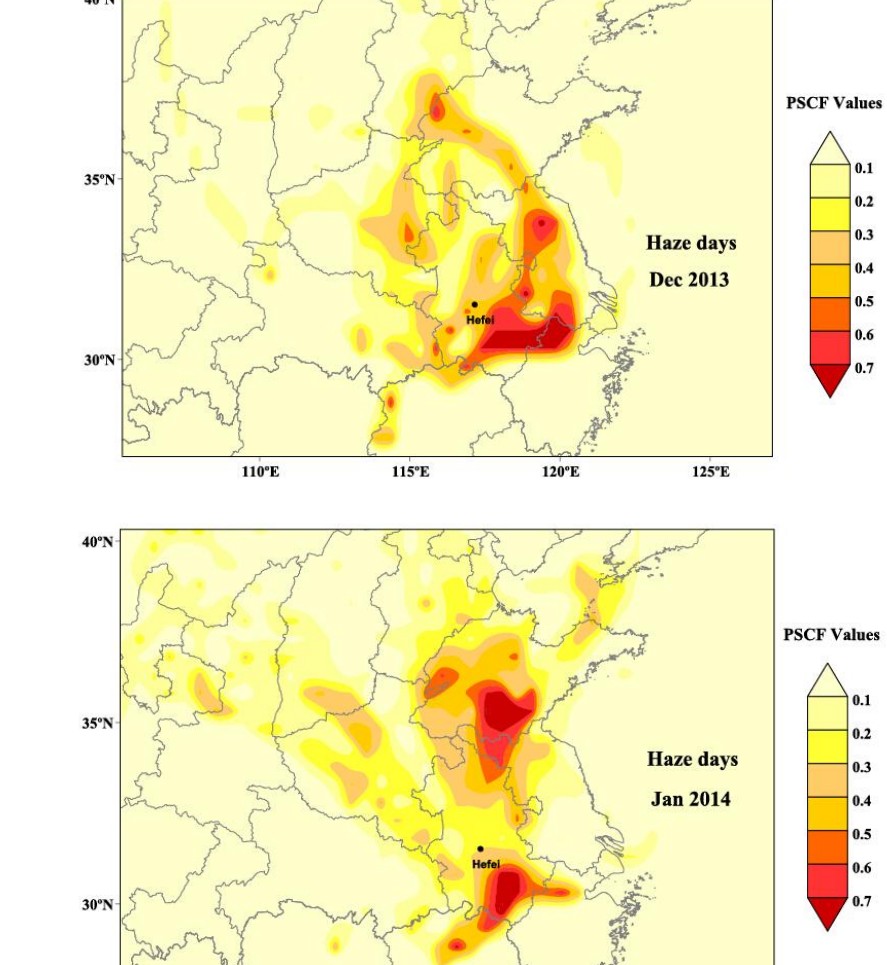



**Fig. 5.** Likely sources areas of GEM during haze days identified by PSCF analysis.













**Fig. 6.** Likely emission sources areas of GEM simulated by PSCF analysis. (A)
overall (from July 2013 to June 2014), (B) spring, (C) summer, (D) autumn, (E)
winter.




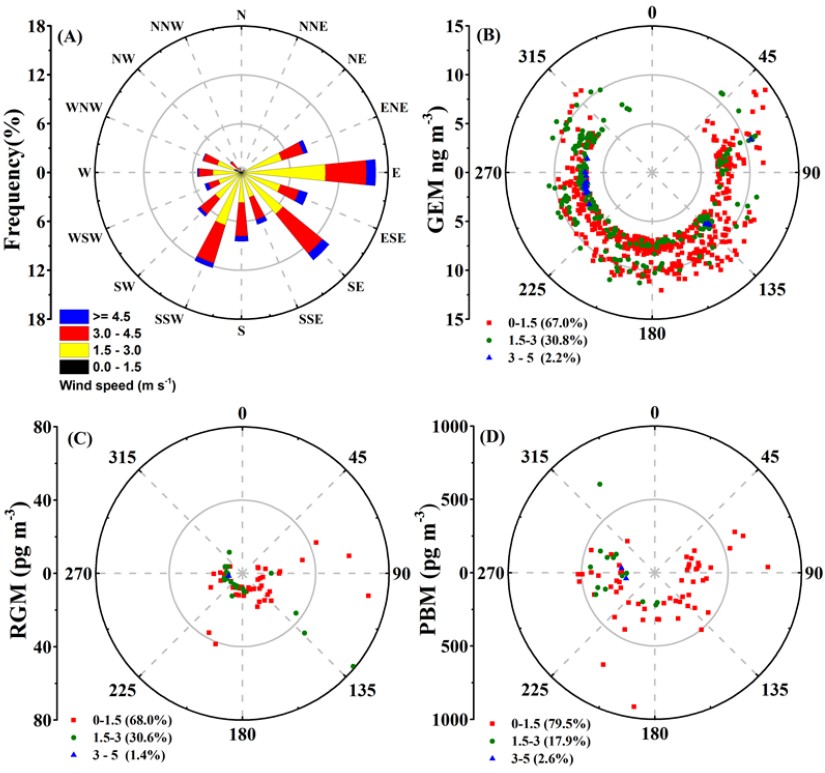



**Fig. 7.** Wind direction and speed at the Science Island Meteorological Station during
the study period. (A) the wind rose for the whole study period; (B), (C) and (D) are
the wind rose diagrams for GEM, RGM and PBM concentrations above the 90th
percentile values, respectively.






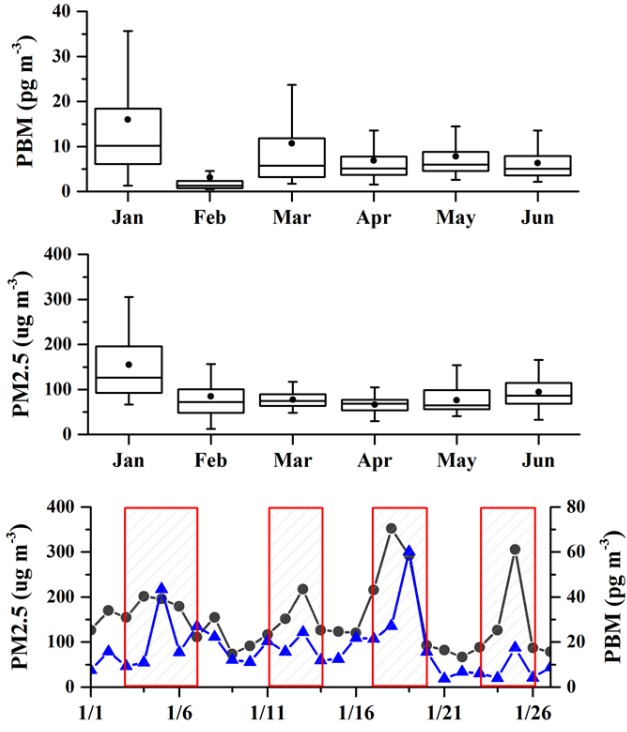


**Fig. 8.** PBM and PM$_{2.5}$ concentrations from January to June, 2014.





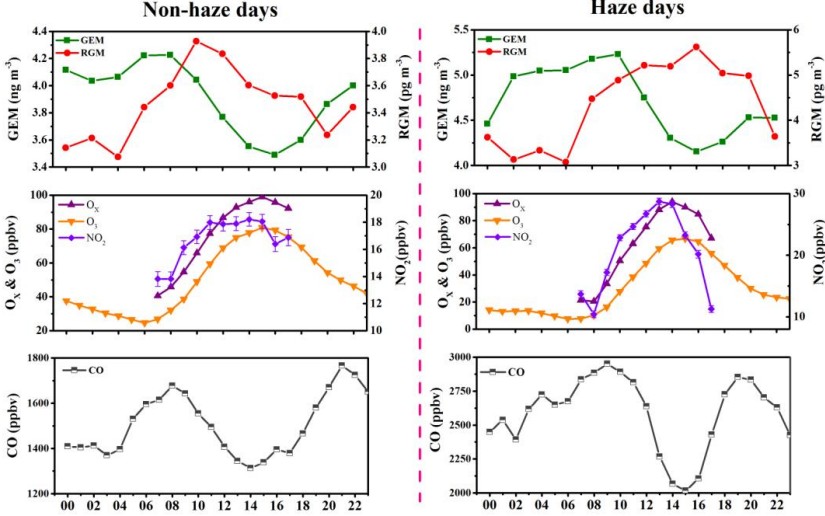


**Fig. 9.** Diurnal variations of GEM, RGM and Gas-Phase Data ($O_X$, $O_3$, $NO_2$ and CO)

concentrations during non-haze and haze days. Notes: bottom is the carbon monoxide

mixing ratio. Middle are the averaged $O_3$, $O_X$ ($O_X = NO_2 + O_3$) and $NO_2$ concentrations.

Top are the hourly averaged GEM and RGM concentrations. The bars for $NO_2$ refer to

the standard deviations.