# Peer review of "Speciated Atmospheric Mercury during Haze and Non-haze Days in an Inland City in China"

_Atmospheric Chemistry and Physics, 2016_

## Referee Comment (RC1) · Anonymous Referee #1 · 28 Jul 2016

The paper presents one year long continuous measurements of mercury species at an urban site, together with measurements of CO, O3, PM2.5 and K+ in daily PM10. NO2 was also derived from DOAS measurements using sun light. The data set is valuable.

The authors interpret the measurements in terms of sources and the diurnal variations of mercury species in terms of chemistry. The discussion of emissions and source areas is questionable. The interpretation of the diurnal variations in terms of chemistry is highly speculative and unconvincing. Taking into account the value of the data, I still recommend a final publication if the authors are able to respond to the comments below and reinterpret the data accordingly:

1. NO2 measurement: MAX-DOAS measures slant column densities which were converted to vertical column densities as described in line 157 to 162. The authors describe how they convert a vertical column density to local NO2 concentrations in the Supplement. They assume homogeneous concentrations within a 500 m thick boundary layer (BL) irrespective of daytime. Constant height of a boundary layer (BL) over a daytime is not realistic and will deliver a false diurnal variation of NO2 concentrations. Neither is a constant height of BL of 500 m applicable to different seasons.

2. Section 4.1: The discussion of the PSCF results is difficult to follow. Figure 5 shows potential source areas of GEM during the haze events in December 2013 and January 2014 but the equivalent figures for non-haze days in December 2013 and January 2014 are shown only in supplementary information. It is their difference which can provide the information about the reason for higher GEM during the hazy days. Dtto about the Figure 6: two seasonal data sets should be presented, one for hazy days and one for non-hazy ones.

3. The discussion of GEM vs CO correlations is deeply flawed. The low GEM/CO slopes are interpreted as if biomass burning were the major source for both GEM and CO in Hefei. To start with GEM/CO slopes represent their emission ratios if a) the background concentrations do not change, b) the emissions remain constant, and c) there is only dilution, no chemistry, on the way from the source to receptor during an event. Using monthly or other "non-event" data would violate at least the condition a) and b). In addition, whatever the sources of GEM might be, in a city of 7 million people and some 1 million of vehicles most of the CO at the site within the city will almost certainly come from local tailpipes rather than from distant isolated fire counts shown in Figure S4. The authors present Figure S3 as additional evidence in favour of biomass burning being the major source. The figure shows correlation between K+ and an Air Quality Index, whose definition is not given in the paper. To be halfway credible, K+ has to correlated with CO. Even if K+ correlated with CO, it still will not prove the biomass burning origin of the mercury. For that the density of the firecounts has to be consistent with results of the PSCF analysis which it evidently is not. In addition remote biomass burning would not yield highest GEM, RGM, and PBM concentration

at the lowest wind speeds – see section 4.2. In summary, the low GEM/CO ratio is characteristic for the emissions of Hefei.

4. Section 4.2: Highest PBM and PM2.5 concentrations in January are most likely due to shallower boundary layer in January than in other months. That is probably meant by "poor diffusion conditions in cold months". The average PBM concentrations in March differ hardly from other months except for January but their spread is larger. I think that the precipitation and the frequency of change of air masses should be also taken into account as driving forces for the PBM vs PM2.5 correlation.

5. Section 4.3: The interpretation of the diurnal variations here is almost certainly wrong. The authors interpret GEM and PBM diurnal variation in terms of changing height of boundary layer and declare that the opposite RGM diurnal variation must be of chemical origin. This must not be and probably is not true. RGM correlates with O3 which is probably not formed in situ but admixed from the free troposphere (FT) as the height of BL increases during the morning. Higher RGM concentrations in FT than in BL have been reported by many researchers. Consequently, the RGM correlation with O3 and its anticorrelation with CO can be viewed as solely a transport phenomenon unrelated to any chemical process. The distinction between a transport and chemical processes is a general problem in the interpretation of diurnal variations. It can only be resolved by careful modeling using measured diurnal variation of the BL height and known concentrations in BL and FT or by using specific tracers for photochemical processes such as peroxynitrates. In this particular case, diurnal variations of GEM, PBM, CO, NOx, etc. emissions due to morning and evening rush hours, working times, etc. additionally complicate the interpretation of the diurnal variations. As mentioned before the diurnal variation of NO2 is also flawed by the assumption of constant height of boundary layer. In summary, the observed diurnal variation can be interpreted solely as a transport phenomenon due to air exchange between BL and FT. As long as the authors cannot rule out the transport hypothesis their chemical interpretation of the diurnal variation and discussion of NO2 kinetics are wishful thinking without any evi-

dential basis.

Editorial comments:

Line 66-67: PBM is not highly surface reactive. "Affinity" might be better than "reactivity".

Line 72: The most recent quoted reference is Pacyna et al. (2006). In 2016 and recent discussions about emissions this seems to be quite obsolete. Dtto line 82. Please quote more recent publications.

Line 322: The sentence is flawed both in content as in grammar. If taken at face value, the text insinuates emissions from power plants being "non-normal" although they represent the largest GEM emissions in most inventories.

Reference at line 584 is incomplete.

Figure 8: Bottom plot: which symbol is PBM and which one PM2.5? The caption of the Figure 8 seems to be inconsistent with the time scale of the bottom plot. The time scale of the bottom plot has not equidistant intervals.

Figure 9: The scales of the y-axes should be same for the haze and non-haze days to facilitate a comparison. E.g. CO mixing ratios are much higher on hazy days.

---

## Referee Comment (RC2) · Anonymous Referee #2 · 16 Sep 2016

This paper clearly describes differences in the atmospheric concentrations of speciated mercury, during haze and non-haze days, at an inland site in central China. The most notable difference was the higher (2.5 times) PBM concentrations on haze days. This paper also used different approaches to identify potential sources and source regions for the atmospheric mercury compounds. In addition, the authors discussed chemical transformation that may have been responsible for the variation in the concentrations of the atmospheric mercury compounds.

I found this paper to be very well written and organized. The data strongly supports their conclusions. However, there are to many figures and I have only a few minor comments, mostly typos that needed to be resolved. I recommend publication in your journal.

[Figure]

Minor issues Line 40: "and" should be placed between "spring, summer".

Line 55 and all other lines: Several years ago, the atmospheric mercury community stopped using the term "reactive gaseous mercury". This term was replaced with "GOM", gaseous oxidized mercury. I suggest you change all of your RGM to GOM.

In fact, you mention "gaseous oxidized mercury" in line 146.

Line 148: change "limits" to "limit".

Line 187: you mentioned an arbitrarily set criterion of 4 ng m-3. I really don't know much about PSCF, but why do you use arbitrary criteria. How will different arbitrary criteria affect your results?

Line 241: "folds" should be fold or two-fold.

Line 296: change "sources region" to "source region".

Line 333: change "condition" to "conditions".

Line 391: remove ( and ) from you reference (Sommar et al. 2001).

Line 454: change "may plays" to "may play"

Line 456: change "greatly" to "great"

Table 2 should be updated to include mercury speciation studies conducted in the US over the past few years. There are several studies in the peer-reviewed literature that can be cited in this table, bring it up to date.

Figure 3. Can you eliminate this figure? These data on already in Table 1.

Figures 4 and 5. Can you merge these two figures into a single figure?

---

## Author Comment (AC1) · 26 Sep 2016

**Response to Anonymous Referee #1**

We thank the reviewer for the constructive suggestions/comments. Below we provide a point-by-point response to individual comments (comments in italics, responses in plain font; page numbers refer to the ACPD version; figures used in the response are labeled as Fig. R1, Fig. R2,… ).

**Comments and suggestions:**

*NO2 measurement: MAX-DOAS measures slant column densities which were converted to vertical column densities as described in line 157 to 162. The authors describe how they convert a vertical column density to local NO2 concentrations in the Supplement. They assume homogeneous concentrations within a 500 m thick boundary layer (BL) irrespective of daytime. Constant height of a boundary layer (BL) over a daytime is not realistic and will deliver a false diurnal variation of NO2 concentrations. Neither is a constant height of BL of 500 m applicable to different seasons.*

**Responses and Revisions:**

Differential Optical Absorption Spectroscopy (DOAS) is used to retrieve $NO_2$ and $O_4$ differential slant column densities (DSCDs) from the measured scattered sunlight spectra (Platt, 1994). In this study, each MAX-DOAS scanning cycle consists of eight elevation viewing angles ($2°$, $3°$, $6°$, $8°$, $10°$, $20°$, $30°$ and $90°$) and lasts about 15 min. The spectra are analyzed using the QDOAS spectral-fitting software suite developed at BIRA-IASB ([http://uv-vis.aeronomie.be/software/QDOAS/](http://uv-vis.aeronomie.be/software/QDOAS/)). Detail information about the spectral fitting for $NO_2$ and $O_4$ is listed in Table S2 (in the revised supplement). As pointed out by the reviewer, MAX-DOAS vertical column densities could not fully represent the $NO_2$ surface concentration. So we used the HEIPRO (Heidelberg Profile, developed by IUP Heidelberg) retrieval algorithm to retrieve $NO_2$ vertical profiles for each MAX-DOAS scanning cycle. The purpose of calculating $NO_2$ profiles is to know the $NO_2$ vertical distribution. More details about $NO_2$ profile retrieval are described in the revised supplement. The $NO_2$ vertical profile

was shown in Fig. R1, indicating that the $NO_2$ is not homogeneously distributed vertically. We agree with the reviewer that converting $NO_2$ $DSCD_S$ to mixing ratios by assuming that the trace gases were homogeneous within the 500 m height of the boundary is not suitable. We took the suggestion from the reviewer and updated the $NO_2$ results in the revision. The retrieval altitude grid is 80 layers of 50m thickness between 0.02 and 3.97 km. Thus, in the revised manuscript we have revised our method by using the surface $NO_2$ concentration (0.02 km) which from the $NO_2$ vertical profile (Fig. R1) to analyze. Due to the large computational requirement, we were not able to complete the calculation of the NO2 vertical profile for the whole year. However, in Figure R1 we showed the result of one such NO2 vertical profile (20th November, 2013), and the DFS and the errors of the retrieval are to determine whether the retrieved method is reasonable or not. As shown in Figure R2, the results suggest that the retrieved $NO_2$ vertical profile is reliable according to the experience of other research (Wang et al., 2014).

[Figure]

**Figure R1**. Example NO$_2$ vertical profiles at six different times (shown on top of each graph as YYYYMMDD hhmmss) from MAX-DOAS measurements in Hefei (20 November, 2013).

[Figure]

**Figure R2.** (a) DFS diurnal cycles corresponding to the NO2 profile retrievals; (b) Errors of NO2 vertical profile retrieval from MAX-DOAS measurements at Hefei (20 November, 2013 at 10:25LT).

**Comments and suggestions:**

*Section 4.1: The discussion of the PSCF results is difficult to follow. Figure 5 shows potential source areas of GEM during the haze events in December 2013 and January 2014 but the equivalent figures for non-haze days in December 2013 and January 2014 are shown only in supplementary information. It is their difference which can provide the information about the reason for higher GEM during the hazy days. Dtto about the Figure 6: two seasonal data sets should be presented, one for hazy days and one for non-hazy ones.*

**Responses and Revisions:**

We have updated and merged PSCF results for potential source areas analysis of GEM in the revised manuscript. Two seasonal data sets are now include, one for haze days and the one for non-haze days. Since the number of haze days accounts only for 5.6% of the total days in spring and summer, we did not provide haze and non-haze PSCF results for spring and summer seasons. As autumn and winter are the prevalent

seasons for haze pollution, one PSCF result for haze days and another for non-haze days are shown for autumn and winter, respectively. We have combined Figure 5 and Figure 6 into a new figure (Figure 4) in the revised manuscript. The updated PSCF results showed that higher GEM concentration was mainly influenced by local emission sources during haze days. For non-haze days, the most important mercury sources to the monitoring site were not only the local emission sources, but also those from the neighboring region of Shandong, Henan and Jiangxi provinces. In summary, the increase of GEM concentration during haze days was mainly caused by local emission.

*Comments and suggestions:*

*The discussion of GEM vs CO correlations is deeply flawed. The low GEM/CO slopes are interpreted as if biomass burning were the major source for both GEM and CO in Hefei. To start with GEM/CO slopes represent their emission ratios if a) the background concentrations do not change, b) the emissions remain constant, and c) there is only dilution, no chemistry, on the way from the source to receptor during an event. Using monthly or other "non-event" data would violate at least the condition a) and b). In addition, whatever the sources of GEM might be, in a city of 7 million people and some 1 million of vehicles most of the CO at the site within the city will almost certainly come from local tailpipes rather than from distant isolated fire counts shown in Figure S4. The authors present Figure S3 as additional evidence in favour of biomass burning being the major source. The figure shows correlation between K+ and an Air Quality Index, whose definition is not given in the paper. To be halfway credible, K+ has to be correlated with CO. Even if K+ correlated with CO, it still will not prove the biomass burning origin of the mercury. For that the density of the fire counts has to be consistent with results of the PSCF analysis which it evidently is not. In addition remote biomass burning would not yield highest GEM, RGM, and PBM concentration at the lowest wind speeds-see section 4.2. In summary, the low GEM/CO ratio is characteristic for the emissions of Hefei.*

**Responses and Revisions:**

Upon further examination of our data, we agree with the reviewer that our original interpretation of the low GEM/CO was not fully supported. Therefore, we have removed Figure S2 (in the ACPD supplement) and revised thoroughly section 4.1 regarding the GEM/CO ratio. The definition of mercury pollution events are not same as haze days in this paper. These mercury pollution episodes were defined as a period with hourly average GEM concentration higher than seasonal average GEM concentration and the duration of elevated hourly GEM concentration lasted for over 10 hours (Kim et al., 2009). We discussed the correlation coefficients and slopes between GEM concentration and CO concentration during pollution events (Table 3 in the revised manuscript). In previous research, the Hg/CO slope and correlation between GEM and CO concentrations has been used to identify long-range transport episodes or local episodes: significant positive correlation for long-range transport episodes and poor correlation for local episodes (Kim et al., 2009). According to the correlations between GEM concentration and CO concentration, the mercury pollution episodes in autumn and winter mainly belong to local episodes. Incomplete combustion like residential coal and biomass burning combustion could lead to a lower Hg/CO ratio. We agree with the reviewer's point. In summary, the low GEM/CO ratio may be characteristic for the local emissions of Hefei.

As for water-soluble potassium ($K^+$) in 24-hr $PM_{10}$ samples, the correlation between $K^+$ and Air Quality Index maybe not reliable. So we did the correlation between $K^+$ and GEM during the 24-hr $PM_{10}$ sampling period (Fig. R3). Although the concentration of water-soluble potassium ($K^+$) in $PM_{10}$ shows a good correlation ($R^2$=0.67) with GEM, due to the small number of compared samples (n=6), so it has great accidental and uncertainty. In addition, most pollutant concentration increased during this heavy pollution episodes (Nov-Dec, 2013). Good correlation might occur between K+ and other pollutant, so it cannot fully prove that GEM come from the emission of biomass burning through good correlation between $K^+$ and GEM. Thus, we agree with the reviewer's comment and shortening this section. We have removed the discussion about $K^+$ and biomass burning altogether in the revised manuscript and supplement.

[Figure]

**Figure R3.** Correlation between water-soluble potassium (K+) and GEM during heavy pollution periods (from 10 Nov to 9 Dec, 2013). Notes: water-soluble potassium ($K^+$) concentrations were analyzed from 24-hr $PM_{10}$ (particulate matter less than 10 μm in diameter) samples, GEM concentrations were the average value during the 24-hr $PM_{10}$ sampling period.

**Comments and suggestions:**

*Section 4.2: Highest PBM and PM2.5 concentrations in January are most likely due to shallower boundary layer in January than in other months. That is probably meant by "poor diffusion conditions in cold months". The average PBM concentrations in March differ hardly from other months except for January but their spread is larger. I think that the precipitation and the frequency of change of air masses should be also taken into account as driving forces for the PBM vs PM2.5 correlation.*

**Responses and Revisions:**

We agree with the reviewer's suggestion. Unfortunately, we did not obtain the precipitation data during our monitoring period, so we were not able to directly examine the influence of precipitation on PBM. This is something we will investigate in our future studies.

Although on average the PBM concentrations in March differ hardly from other months (April-June), they fluctuated much greatly in March when compared to other months. We have re-examined the wind rose diagrams for March and April (see Fig.

R4). The prevailing wind direction in March indeed varied much greatly than in April. Thus, the larger fluctuation of PBM in March might be related to the frequency of change in wind direction. We have thus removed our original interpretation that "higher temperatures in the warmer months do not favor mercury adsorption", and replaced it with reference to changes in wind direction.

[Figure]

Figure R4. The wind rose diagrams for (a) March and (b) April.

**Comments and suggestions:**

*Section 4.3: The interpretation of the diurnal variations here is almost certainly wrong. The authors interpret GEM and PBM diurnal variation in terms of changing height of boundary layer and declare that the opposite RGM diurnal variation must be of chemical origin. This must not be and probably is not true. RGM correlates with O3 which is probably not formed in situ but admixed from the free troposphere (FT) as the height of BL increases during the morning. Higher RGM concentrations in FT than in BL have been reported by many researchers. Consequently, the RGM correlation with O3 and its anticorrelation with CO can be viewed as solely a transport phenomenon unrelated to any chemical process. The distinction between a transport and chemical processes is a general problem in the interpretation of diurnal variations. It can only be resolved by careful modeling using measured diurnal variation of the BL height and known concentrations in BL and FT or by using specific tracers for photochemical processes such as peroxynitrites. In this particular case, diurnal variations of GEM, PBM, CO, NOx, etc. emissions due to morning and*

*evening rush hours, working times, etc. additionally complicate the interpretation of the diurnal variations. As mentioned before the diurnal variation of NO2 is also flawed by the assumption of constant height of boundary layer. In summary, the observed diurnal variation can be interpreted solely as a transport phenomenon due to air exchange between BL and FT. As long as the authors cannot rule out the transport hypothesis their chemical interpretation of the diurnal variation and discussion of NO2 kinetics are wishful thinking without any evidential basis.*

**Responses and Revisions:**

We agree with the reviewer that resolving transport and reaction processes of RGM is not straightforward; the fact that we did not measure specific photochemical tracers such as peroxynitrites did not help. Two processes can affect the RGM concentrations in the boundary layer air. The first is due to transport of RGM from the free troposphere (FT). Diurnal variations of GEM, RGM, $O_3$ and CO concentrations during non-haze and haze days are shown in Fig. R5 (Figure 7 in the revised manuscript). For both non-haze and haze days, RGM concentrations remained at a relatively constant level during night, and then increased suddenly prior to the sunrise. We agree with the reviewer that such enhancement of RGM in early morning might can be due, at least in part, to its transport from the free troposphere as the height of BL increases. In summary, the observed RGM diurnal variation can be interpreted as a transport phenomenon due to air exchange between BL and FT.

In addition, in situ photochemical oxidation of GEM could also increase the concentration of RGM during daytime. To determine the relative importance of FT transport and in situ photochemical oxidation, we examined the relationship between RGM and the changes in the height of the atmospheric boundary layer and the odd oxygen ($O_X = O_3 + NO_2$) concentrations. Although we did not measure peroxynitritesin this study, we believe the concentration of odd oxygen ($O_X = O_3 + NO_2$ which is produced from the reaction between O3 and NO) can be used as a tracer of the extent of photochemical processing in the urban atmosphere. Since $NO_2$ concentrations from MAX-DOAS were only available during daytime, we could only use $O_X$ to be a

indicator for daytime GEM oxidation. As per our manuscript, we selected 20th November 2013 as a case study to probe the importance of photochemical processes. Both RGM and $O_X$ reached higher concentrations from 12:00 to 16:00, along with the lowest value of GEM. The height of atmospheric boundary layer changed very little over this period (12:00-16:00, see Fig. R6). This simple comparison suggests that the transport of FT RGM might be limited and that at least some of the RGM were formed from in situ oxidation of GEM. We further investiagted the potential mechanism of the GEM oxidation to GOM.

[Figure]

**Figure R5.** Diurnal variations of GEM, RGM, $O_3$, and CO concentrations during non-haze and haze days.

[Figure]

**Figure R6.** A case study of diurnal variations of GEM, RGM, $O_X$, and $NO_2$ at Hefei (20th November, 2013, left panel). The right panel shows the retrieved aerosol extinction profile on the same day; the black line represents the height of the atmospheric boundary layer.

*Comments and suggestions:*

*Line 66-67: PBM is not highly surface reactive. "Affinity" might be better than "reactivity".*

**Responses and Revisions:**

Corrected.

*Comments and suggestions:*

*Line 72: The most recent quoted reference is Pacyna et al. (2006). In 2016 and recent discussions about emissions this seems to be quite obsolete. Dtto line 82. Please quote more recent publications.*

**Responses and Revisions:**

We have updated the section by quoting two recent publications ((Pacyna et al., 2010) and (Zhang et al., 2015)).

*Comments and suggestions:*

*Line 322: The sentence is flawed both in content as in grammar. If taken at face value, the text insinuates emissions from power plants being "non-normal" although they represent the largest GEM emissions in most inventories. Reference at line 584 is incomplete.*

**Responses and Revisions:**

The sentence in Line 322 has been modified according to the revision in the revised manuscript (second paragraph of section 4.1). We modified this sentence as follows: "GEM and CO often share similar anthropogenic emission sources, such as industrial coal combustion, domestic coal combustion, iron and steel production and cement production (Wu et al., 2006;Wang et al., 2005). However, they also have their own sources. For instance, power plants and nonferrous metal smelters emit mercury but

hardly any CO, while most of CO originates from vehicles which are not a major emitter for mercury."

We also corrected the reference at line 584 (Hu et al., 2014).

**Comments and suggestions:**

*Figure 8: Bottom plot: which symbol is PBM and which one PM2.5? The caption of the Figure 8 seems to be inconsistent with the time scale of the bottom plot. The time scale of the bottom plot has not equidistant intervals.*

**Responses and Revisions:**

We have rearranged the figures, added the symbols of PBM and $PM_{2.5}$ and updated the caption. The time scale of the bottom plot has corrected and had equidistant intervals.

**Comments and suggestions:**

*Figure 9: The scales of the y-axes should be same for the haze and non-haze days to facilitate a comparison. E.g. CO mixing ratios are much higher on hazy days.*

**Responses and Revisions:**

We have redrawn the figures so that the scales of the y-axes are same for the haze and non-haze days.

**References**

Kim, S. H., Han, Y. J., Holsen, T. M., and Yi, S. M.: Characteristics of atmospheric speciated mercury concentrations (TGM, Hg(II) and Hg(p)) in Seoul, Korea, Atmospheric Environment, 43, 3267-3274, doi:10.1016/j.atmosenv.2009.02.038, 2009.

Pacyna, E. G., Pacyna, J., Sundseth, K., Munthe, J., Kindbom, K., Wilson, S., Steenhuisen, F., and Maxson, P.: Global emission of mercury to the atmosphere from anthropogenic sources in 2005 and projections to 2020, Atmospheric Environment, 44, 2487-2499, 2010.

Platt, U.: Differential optical absorption spectroscopy (DOAS), Air monitoring by spectroscopic technique, 127, 27-84, 1994.

Wang, L., Zhang, Q., Hao, J., and He, K.: Anthropogenic CO emission inventory of Mainland China, Acta Scientiae Circumstantiae, 25, 1580-1585, 2005.

Wang, T., Hendrick, F., Wang, P., Tang, G., Clémer, K., Yu, H., Fayt, C., Hermans, C., Gielen, C., and Müller, J.-F.: Evaluation of tropospheric $SO_2$ retrieved from MAX-DOAS measurements in Xianghe, China, Atmospheric Chemistry and Physics, 14, 11149-11164, 2014.

Wu, Y., Wang, S., Streets, D. G., Hao, J., Chan, M., and Jiang, J.: Trends in anthropogenic mercury emissions in China from 1995 to 2003, Environmental science & technology, 40, 5312-5318, 2006.

Zhang, L., Wang, S., Wang, L., Wu, Y., Duan, L., Wu, Q., Wang, F., Yang, M., Yang, H., and Hao, J.: Updated Emission Inventories for Speciated Atmospheric Mercury from Anthropogenic Sources in China, Environmental science & technology, 49, 3185-3194, 2015.

---

## Author Comment (AC2) · 26 Sep 2016

The comment was uploaded in the form of a supplement:
http://www.atmos-chem-phys-discuss.net/acp-2016-467/acp-2016-467-AC2-
supplement.pdf

---

## Author Comment (AC3) · 26 Sep 2016

**Response to Anonymous Referee #2**

We thank the reviewer for the constructive suggestions/comments. Below we provide a point-by-point response to individual comments (comments in italics, responses in plain font; page numbers refer to the ACPD version; figures used in the response are labeled as Fig. R1, Fig. R2,… ).

**Comments and suggestions:**

*Minor issues Line 40: "and" should be placed between "spring, summer".*

**Responses and Revisions:**

Corrected.

**Comments and suggestions:**

*Line 55 and all other lines: Several years ago, the atmospheric mercury community stopped using the term "reactive gaseous mercury". This term was replaced with "GOM", gaseous oxidized mercury. I suggest you change all of your RGM to GOM.*

**Responses and Revisions:**

Agreed and corrections have been made in the text and figures.

**Comments and suggestions:**

*Line 148: change "limits" to "limit".*

**Responses and Revisions:**

Corrected.

**Comments and suggestions:**

*Line 187: you mentioned an arbitrarily set criterion of 4 ng m-3. I really don't know much about PSCF, but why do you use arbitrary criteria. How will different arbitrary criteria affect your results?*

**Responses and Revisions:**

The "arbitrarily set criterion" was used in the definition of $M_{ij}$ in the PSCF analysis (see TrajStat_Help_v1.2). The definition of $M_{ij}$ as follows: The number of endpoints for the same cell having arrival times at the sampling site corresponding to pollutant

concentration higher than an arbitrarily set criterion is defined to be $M_{ij}$. In this study, pollutant concentration refers to atmospheric mercury concentration (GEM concentration). The words "arbitrarily set criterion" were used in PSCF method introduction and could also be found in (Fu et al., 2012). However, in the actual operation, we use a fixed GEM value as criterion. In this study, mean GEM concentration of 4 ng m$^{-3}$ during the whole study period was used as fixed criterion (refer to Fu et al., 2012). We split the sentence in Lines 192-195 (ACPD version) to describe clearly in the revised manuscript.

**Comments and suggestions:**

*Line 241: "folds" should be fold or two-fold.*

**Responses and Revisions:**

Corrected

**Comments and suggestions:**

*Line 296: change "sources region" to "source region".*

**Responses and Revisions:**

Corrected.

**Comments and suggestions:**

*Line 333: change "condition" to "conditions".*

**Responses and Revisions:**

Corrected.

**Comments and suggestions:**

*Line 391: remove (and) from you reference (Sommar et al. 2001).*

**Responses and Revisions:**

Corrected

**Comments and suggestions:**

*Line 454: change "may plays" to "may play"*

**Responses and Revisions:**

Corrected.

**Comments and suggestions:**

*Line 456: change "greatly" to "great"*

**Responses and Revisions:**

Corrected.

**Comments and suggestions:**

*Table 2 should be updated to include mercury speciation studies conducted in the US over the past few years. There are several studies in the peer-reviewed literature that can be cited in this table, bring it up to date.*

**Responses and Revisions:**

Agreed. We have added (Peterson et al., 2012) and (Ren et al., 2016) in Table 2.

**Comments and suggestions:**

*Figure 3. Can you eliminate this figure? These data on already in Table 1.*

**Responses and Revisions:**

We assume the reviewer meant Figure 4 in the ACPD version. The only overlap of data between this figure and Table 1 are GEM, RGM and PBM mean concentrations. This figure also provides further information such as various percentiles. We agree with the reviewer's comment and move this figure into the supplement.

**Comments and suggestions:**

*Figures 4 and 5. Can you merge these two figures into a single figure?*

**Responses and Revisions:**

We assume the reviewer meant Figures 5 and 6. We have merged PSCF results for potential source areas analysis of GEM in the revised manuscript. Two seasonal data sets have presented, one for haze days and another one for non-haze days.

**References**

Fu, X. W., Feng, X., Shang, L. H., Wang, S. F., and Zhang, H.: Two years of measurements of atmospheric total gaseous mercury (TGM) at a remote site in Mt. Changbai area, Northeastern China, Atmospheric Chemistry and Physics, 12, 4215-4226, doi:10.5194/acp-12-4215-2012, 2012.

Peterson, C., Alishahi, M., and Gustin, M. S.: Testing the use of passive sampling systems for understanding air mercury concentrations and dry deposition across Florida, USA, Science of the Total Environment, 424, 297-307, 2012.

Ren, X., Luke, W. T., Kelley, P., Cohen, M. D., Artz, R., Olson, M. L., Schmeltz, D., Goldberg, D. L., Ring, A., and Mazzuca, G. M.: Atmospheric mercury measurements at a suburban site in the Mid-Atlantic United States: Inter-annual, seasonal and diurnal variations and source-receptor relationships, Atmospheric Environment, 2016.

---

## Author Response (AR2)

**Response to Editor Comments**

We thank the editor for the constructive suggestions/comments. Below we provide a point-by-point response to individual comments (comments in italics, responses in plain font; figures used in the response are labeled as Fig. R1, Fig. R2,… ).

*Comments and suggestions:*

*You and your coauthors have addressed most of the referees' comments satisfactorily. However, I remain concerned about your boundary layer depth estimates based solely on MAX-DOAS data. Your analysis would be strengthened by independent BL depth estimates based on, e.g., radiosonde data from locations not too distant from Hefei. I believe that there are data available from Nanjing and Shanghai covering your observation period. Although there are only 2 observations per day, they are at times (at 0 and 12 UTC) that could yield some information about diurnal variability. This manuscript on BL depth climatology in China that is also currently in review might be worth looking at as well: http://www.atmos-chem-phys-discuss.net/acp-2016-564/.*

**Responses and Revisions:**

We accept your suggestions about boundary layer (BL) depth and we have calculated BL height from ground meteorological observation data. As the Editor pointed out, Radiosonde data is a good choice but there are only 2 observations per day (at 8 and 20 Local time), so it might be difficult to explain the daytime variation of BL height for the case study on 20 Nov, 2013 (see section 4.3). Fortunately, we have found other meteorological data to calculate BL height in the revised ms. We used the national standard method (GB/T13201-91, referred to as GB method) of China, to calculate the height of atmospheric boundary layer. The GB method considers the thermal condition of the surface layer greatly depends on the heating and cooling degrees of the ground.

In the revision, the entire year (from July 2013 to June 2014) of the surface meteorological data acquired by the China Meteorological Administration (CMA) data network was used to calculate the boundary layer height (BLK) in Hefei. Detailed information about the GB method and calculation formula can be found in the supplement. The calculated BL height is at a 3-hours resolution (at 2, 5, 8, 11, 14, 17, 20

and 23 Local time).

Diurnal variation of boundary layer height (BLK) on non-haze and haze days is shown in Fig. R1. As we suggested in the original ms, BLH is indeed low in the morning and night, and high during the daytime. These results further support our original interpretation that the diurnal variation of GEM and PBM could be related to changes in the height of urban boundary layer (section 4.2, second paragraph).

[Figure]

**Fig. R1.** Diurnal variation of boundary layer height (BLK) on non-haze and haze days in Hefei. Notes: the atmospheric boundary layer height data were calculated by the

GB method.

Figure R2 (Figure S4 in the revised supplement) showed the boundary layer height from the retrieved aerosol extinction profile (left). In order to strengthening boundary layer height results, boundary layer height calculated from CMA meteorological observation data in Hefei for 20 November, 2013 was shown in Figure R2 (right). The height of the atmospheric boundary layer changed very little (less than 0.1 km) at noon.

[Figure]

**Fig. R2.** Retrieved aerosol extinction profile for a case study on 20 November,

2013. The black line represents the height of atmospheric boundary layer during daytime (left). The atmospheric boundary layer height (BLK) was calculated from

China Meteorological Administration (CMA) meteorological observation data on 20

November, 2013 (right).

*Comments and suggestions:*

*One other (and relatively minor) point on content: It would be helpful to add p*

*values for the correlations summarized in Table 3.*

**Responses and Revisions:**

Agreed and p values for the correlations have been added in Table R1 (Table 3 in the revised manuscript). The p values for three events (#2, 4 and 5) are larger than 0.05, the slopes are not significant correlation (at the 0.05 level). We have removed these events (#2, 4 and 5) in Table R1 (Table 3).

Table R1. Coefficients of determination and slopes between GEM concentration and CO concentration during atmospheric mercury pollution events (* p<0.01).

| Event | Start Time (UTC + 8 hr) | End Time (UTC + 8 hr) | Duration (h) | GEM (ng m$^{-3}$) | CO (ppbv) | GEM/CO (slope, ng m$^{-3}$ ppbv$^{-1}$) | $R^2$ | p |
|---|---|---|---|---|---|---|---|---|
| 1 | 2013/11/21 03:00 | 2013/11/22 02:00 | 23 | 8.37 ±2.42 | 4481.6± 717.3 | 0.001 8 | 0.29* | 0.00 726 |
| 2 | 2013/12/03 20:00 | 2013/12/04 09:00 | 13 | 7.51 ±0.67 | 5270.0± 744.5 | 0.000 1 | 0.02* | 0.651 15 |
| 3 | 2013/12/07 04:00 | 2013/12/09 04:00 | 48 | 9.21 ±1.16 | 5943.8± 1394.1 | 0.000 4 | 0.23* | 5.82 259E-4 |
| 4 | 2013/12/19 09:00 | 2013/12/20 09:00 | 24 | 4.35 ±0.17 | 3907.6± 353.0 | 0.000 2 | 0.03* | 0.74 582 |
| 5 | 2013/12/24 19:00 | 2013/12/25 15:00 | 20 | 5.58 ±0.94 | 4930.8± 919.7 | 0.001 2 | 0.01* | 0.62 229 |
| 6 | 2014/01/17 22:00 | 2014/01/19 13:00 | 39 | 5.80 ±0.83 | 5746.3± 1626.9 | 0.000 3 | 0.28* | 4.69 294E-4 |
| 7 | 2014/01/25 02:00 | 2014/01/25 22:00 | 20 | 6.03 ±0.50 | 8797.9± 2244.3 | 0.000 2 | 0.59* | 6.91 209E-5 |
| 8 | 2014/03/16 05:00 | 2014/03/16 20:00 | 15 | 4.46 ±0.47 | 2261.7± 440.2 | 0.001 0 | 0.79* | 4.03 625E-6 |
| 9 | 2014/03/17 06:00 | 2014/03/18 12:00 | 30 | 8.85 ±2.46 | 2697.1± 590.3 | 0.003 0 | 0.51* | 6.98 676E-6 |
| 10 | 2014/05/21 00:00 | 2014/05/21 11:00 | 11 | 5.74 ±0.94 | 3676.7± 1690.0 | 0.005 0 | 0.79* | 9.81 815E-5 |

***Comments and suggestions:***

*Although the English is quite good, it is not up to ACP standards. There are still*
*many grammatical and punctuation errors and a few instances of poor word usage. I*
*recommend that you ask a colleague who is a native English speaker to help you*
*remedy this.*

**Responses and Revisions:**

Many thanks! We have thoroughly polished the English in the revised manuscript.

[revised manuscript text omitted]